# Effect of Fermented Sarco Oyster (*Crassostrea gigas*) Extract on Muscle Strength Enhancement in Postmenopausal Females: A Randomized, Double-Blind, Placebo-Controlled Trial

**DOI:** 10.3390/ijerph192416450

**Published:** 2022-12-08

**Authors:** Kyoung-Min Rheu, Bae-Jin Lee, Woo-Hyeon Son, Dong-Seok Kim, Hyun-Tae Park, Min-Seong Ha, Byong-Hak Gong, Byeong-Hwan Jeon

**Affiliations:** 1Marine Bioprocess Co., Ltd., Busan 46048, Republic of Korea; kmin.rheu@gmail.com (K.-M.R.); hansola82@hanmail.net (B.-J.L.); 2Institute of Convergence Bio-Health, Dong-A University, Busan 49236, Republic of Korea; physical365@gmail.com; 3Department of Sports and Health Science, Kyungsung University, Busan 48434, Republic of Korea; dongseok@me.com; 4Graduate School of Health Care and Sciences, College of Health Science, Dong-A University, Busan 49236, Republic of Korea; htpark@dau.ac.kr; 5Department of Sports Culture, Dongguk University, Seoul 04620, Republic of Korea; haminseong@dgu.ac.kr; 6Korea Sports Culture Association, Busan 04420, Republic of Korea; gongbk@hanmail.net

**Keywords:** fermented oyster, γ-aminobutyric acid, insulin growth factor-1, muscle strength, postmenopausal female, dietary supplements, sarcopenia

## Abstract

A randomized controlled trial (RCT) was conducted to evaluate the effect of fermented sarco oysters (FSO) on muscle strength in postmenopausal females with low muscle mass. Fifty-two female participants were randomly divided into the experiment group (EG) or control group (CG). For 12 weeks, the EG was subjected to 1000 mg of FSO extract daily while the CG consumed the placebo extract. The muscle extension and flexion at an angular velocity of 60°/s and with respect to grip strength, body composition, and muscle growth-related blood factors were measured at the baseline and after the trial. The difference in the quadriceps muscle extension at an angular velocity of 60°/s, grip strength on both the left and right side, and insulin-like growth factor-1(IGF-1) between groups were significantly higher in the EG compared with the CG. However, no differences were found in body composition, blood pyruvate, lactate, or high-sensitivity C-reactive protein (hsCRP) concentration between the two groups. In conclusion, FSO supplements may improve muscle strength in postmenopausal females with relatively reduced muscle strength without a change in muscle mass.

## 1. Introduction

Sarcopenia is a degenerative disease with the involuntary loss of skeletal muscle mass, strength, and dysfunction with age. As a type of senile disease, sarcopenia can have profound consequences on both health and quality of life, including reduced muscle function, restricted mobility, disability and frailty, and rheumatoid arthritis (RA) [1,2,3]. The pathophysiology of sarcopenia involves many factors, including low levels of physical activity, low intake of protein, and age-related changes in the hormonal factors [3]. Studies have shown that a decline in the levels of growth hormone (GH) and insulin-like growth factor 1 (IGF-1), the major mediator of GH action, has a strong association with the reduction in muscle function [4]. Both GH and IGF-1 are important factors responsible for the anabolic effects on skeletal muscle tissue. IGF-1 plays a key role and has been shown to stimulate muscle cell proliferation and act as a regulator for both muscle protein synthesis and degradation [5]. Studies have proven that γ-aminobutyric acid (GABA) can directly or indirectly stimulate GH release and IGF-1 expression [6,7].

Menopause occurs due to a decline in estrogen level, which leads to a decrease in bone density, muscles mass, and muscle strength; this is known as sarcopenia, which frequently occurs in postmenopausal women [3]. A previous statistic reported that 10–40% of postmenopausal females had the prevalence of sarcopenia [8]. Hence, preventing sarcopenia can prevent physical impairment and related disabilities and thus enhance the quality of life of people. At present, resistance training has been an effective method for attenuating muscle loss and strength in postmenopausal women [3]; an effective method for the prevention and treatment of sarcopenia is needed.

The *Crassostrea gigas* oyster is one of the world’s most industrially important seafood due to its high sustainability and due it being a great source of protein, vitamins, minerals (such as zinc, iron, calcium, and selenium), and, especially, a large amount of glutamic acid, a precursor of GABA [9,10]. GABA is the main inhibitory neurotransmitter in the central nervous system, and by blocking the chemical messages and delaying the stimulation of the nerve cells, it can affect several functions in the body, including muscle health. Previous studies have shown that enriched GABA contents after being fermented by *Lactobacillus brevis* BJ-20 (*L. brevis* BJ-20) had a positive effect on height growth [11], bone formation [12], and muscle endurance [13].

Thus, fermented sarco oyster (FSO) extract was expected to improve the management of sarcopenia. In our pilot study, an oral intake of 1000 mg of FSO supplementation for 8 weeks increased the muscle endurance and concentration of IGF-1 in the experimental group (EG). Based on the previous studies, FSO may have a beneficial effect on the improvement of skeletal muscle mass and strength in postmenopausal women.

Therefore, this clinical trial was undertaken to evaluate the effect of FSO on muscle strength in postmenopausal females over 12 weeks.

## 2. Materials and Methods

### 2.1. Study Design and Ethics

This study was conducted at Kyungsung University and was designed as a randomized, placebo-controlled, double-blind clinical trial. All participants were provided a signed consent form prior to the study, and their responses were treated confidentially and anonymously.

### 2.2. Participant Eligibility

Participants aged 65 years or older, female, with a body mass index (BMI) between 18.5 and 30.0 kg/m^2^, and with a relatively low skeletal muscles mass (<110% of the standard lean mass) were eligible for the study. Participants with abnormal liver or renal function (aspartate aminotransferase (AST) or alanine aminotransferase (ALT) ≥ 60 IU/L, creatinine level ≥ 1.2 mg/dL, urinalysis dipstick reading of ≥2+), uncontrolled hypertension (blood pressure (BP) ≥ 160/100 mmHg), uncontrolled hyperthyroidism or hypothyroidism, uncontrolled diabetes (fasting glucose level ≥ 160 mg/dL), a history of gastrectomy, mental disorder, known allergies, addiction to alcohol or drugs, and those with a notable cardiovascular disease or central bone fracture within the past 6 months were excluded. Fifty-two females participated in our study.

### 2.3. Randomization

A double-blind, randomized placebo-controlled trial was conducted. The subjects were assigned a 1:1 allocation ratio to the experimental group (EG) and control group (CG) using the block randomization method. The block randomization method in SPSS Statistics for Windows, Version 22.0 (IBM Corp., Armonk, NY, USA) was used. Because the size of the block was arbitrarily set by a third-party statistician, the size and number of blocks were unknown to all investigators. The random list was offered in a sealed, non-permeable envelope, and all participants, investigators, and assessors were blinded to the treatment allocation until the end of the study.

The two study populations were analyzed as the intention-to-treat (ITT) and per-protocol (PP) populations. All subjects with at least one primary or secondary efficacy endpoint datum obtained after the first visit were included in the ITT analysis. Subjects who consumed more than 80% of the total without skipping more than 5 consecutive days of the test product were included in the PP analysis. Accordingly, the total number of subjects in the ITT analysis group was 52 (EG *n* = 26, CG *n* = 26), and the total number of subjects in the PP analysis group was 46 (EG *n* = 23, CG *n* = 23), excluding 6 dropouts. 

Every participant was asked to visit the center four times during the study (visit 1: for screening; visit 2: for randomized supplement distribution; visit 3: 6 weeks after the intervention; visit 4: 12 weeks after for the final observation).

### 2.4. Intervention

The FSO extract and placebo were manufactured by Marine Bioprocess Co. Ltd. (Busan, Republic of Korea). The EG participants were given 1000 mg of FSO/day orally in the form of a 250 mg capsule. Four capsules were taken 30 min after a meal with water for 12 weeks. The placebo capsule was identical to the FSO capsule in appearance and was filled with dextrin. In a previous animal study, no toxicity was observed with 100 mg/kg and 200 mg/kg admission of FSO for 28 days [14]. Therefore, a safe and effective dose of 200 mg/kg was converted to the dose appropriate for a 60 kg adult human based on the body surface area and was calculated as 960 mg/day. In the previous preliminary clinical study, the established safe dose of FSO extract was 1000 mg/day [15]. Therefore, a dose of 1000 mg/day was selected as the final dose for the convenience of intake. The content of FSO and placebo capsules are presented in Table 1. The composition of FSO used in this study is equivalent to that of the previous study.

### 2.5. Procedures

#### 2.5.1. Efficacy Measurement

The primary efficacy endpoint was the change in quadriceps muscle strength measured by the extension and flexion motions at angular velocities at 60°/s, which represented the muscle function at 12 weeks of FSO or placebo use. The secondary efficacy endpoints were the changes in the handgrip strength, body composition, and creatinine, pyruvate, lactate, IGF-1, and high-sensitivity C-reactive protein (hsCRP) concentrations [16], along with the quality of life questionnaire (Euro-Qol-5D (EQ-5D)) score [17].

#### 2.5.2. Biomarker Measurement

Blood samples from all participants were collected after a 12-h fasting at baseline and at 12 weeks to explore the potential effect of the FSO extract. Pyruvate concentrations were measured by an enzymatic assay kit (Determiner PA, Kyowa Medex Co. Ltd., Milano, Italy) on a URIT-800 chemistry analyzer (URIT, Guanxi, China). Lactate concentrations were measured by a colorimetry method kit (Lactate Gen.2., Roche, Manheim, Germany), and hsCRP concentrations were measured by a turbidmetric immunoassay (TIA) kit (CRP4, Roche, Manheim, Germany) on a Cobas C702 chemistry analyzer (Roche, Manheim, Germany). IGF-1 concentrations were measured by an electrochemiluminescence immunoassay (ECLIA) kit (Elecsys IFG-1, Roche, Manheim, Germany) on a Cobas-e801 chemistry analyzer (Roche, Manheim, Germany).

#### 2.5.3. Body Composition Analysis

The body composition analysis was determined by using a multi-frequency bioelectrical impedance analysis (BIA) (InbodyS10, Inbody Co., Ltd., Gangnam, Seoul, Republic of Korea) in the standing posture as per the guidelines of the manufacturer. The appendicular skeletal mass (ASM) was defined as the sum of the muscle masses of the 4 limbs. In this study, the appendicular skeletal mass index (ASMI) was determined by using height squared (ASM/height²) and weight (ASM/weight × 100). All participants were asked to remove their shoes and stand on the platform for 10 minutes prior to the measurements. The measurements were obtained at baseline and after 12 weeks.

#### 2.5.4. Dietary Intake and Physical Activity Assessment

The participants were asked to maintain their usual diet and were required to walk from 30 min to 1 h per day and > 3 times/week during the 12-week trial period. A walking log was used to assess the walking activity. At baseline and after 12 weeks, the participants were asked to answer a questionnaire on physical activities and dietary intake. The International Physical Activity Questionnaire (IPAQ) was used to assess physical activity [18] and the 24-h dietary recall method was used to assess food intake.

### 2.6. Sample Size Calculation

The sample size was calculated using G*Power 3.1.9.2. [19]. Previous nutritional intervention studies with a similar outcome variable (angular velocity of 60°/s measured by the Biodex) were used to estimate the sample size. We derived a small-to-medium effect size of intervention from these studies (Cohen’s *f* = 0.29). Thus, with a statistical power of 0.80, probability level of 0.05, and effect size of 0.20, a sample size of 45 participants was deemed necessary to achieve sufficient power. Considering a 30% dropout, we recruited a total of 52 participants and randomly allocated them into two groups (*n* = 26 in each group): (1) intervention and (ii) control. Figure 1 describes the study design, with participant recruitment and exclusion criteria. The study procedures were approved by the Research Ethics Committee of Kyungsung University (IRB NO. KSU-21-03-004, 17 June 2021).

### 2.7. Safety Assement

For safety assessment, participants were queried about adverse events and tested for abnormalities at the baseline and 12th week. A paired *t*-test was performed for continuous data, and a chi–square test was performed for categorical data. For the ITT population (test group 26, control group 26), with an intake of the experiment supplement more than once, the safety was evaluated by evaluating the biological signs and by conducting clinical laboratory tests, using adverse reactions and laboratory tests as variables during the test period. The clinical laboratory test results were compared between the groups for the amount of change compared with the baseline.

### 2.8. Statistical Analysis

We analyzed the results based on the PP analysis. For all statistical analysis, SPSS Version 22 (SPSS Statistics for Windows Version 22.0, IBM Corp., Armonk, NY, USA) was used, and a two-sided test was performed under a significance level of 0.05. Intergroup comparisons of the baseline characteristics were performed using a chi-square test for the categorical variables or using Fisher’s exact test for non-parametric categorical variables; an independent *t*-test was performed for continuous variables, and the Mann–Whitney U test was performed for non-parametric categorical variables. As for intragroup comparison, we performed a paired *t*-test or Wilcoxon’s rank-sum test for the non-parametric variables. The differences between each endpoint before and after the treatments were analyzed.

## 3. Results

### 3.1. Baseline Demogrpahic Characteristics

We screened 58 participants aged between 65 and 80 years old, and 52 participants were determined to be eligible for the study. The participants were separated into two groups: EG (*n* = 26) or CG (*n* = 26). Six participants (EG: *n* = 3, CG: *n* = 3) withdrew consent for personal reasons and were excluded from the study. Hence, the ITT and PP populations were 52 and 46, respectively. Overall, 46 female subjects completed the trial. The flow of participants is depicted in a CONSORT (Consolidated Standards of Reporting Trials) conform diagram [5] (Figure 1). The demographic and clinical data had no significant difference at the baseline between the two groups (Table 2).

### 3.2. Primary Outcome

The primary efficacy analysis demonstrated that after 12 weeks, there was a significant increase in the right (dominant) quadriceps muscle strength (60°/s, extension) in the EG (*p* < 0.02) compared with the CG, based on the PP analysis (Table 3). Furthermore, the ITT analysis also showed an increase in right quadriceps muscle strength (60°/s, extension) in the EG (*p* < 0.01) (Table 4), (See Appendix A).

### 3.3. Secondary Outcome

In the secondary efficacy analysis, the grip force strength in both the left (*p* < 0.04) and right (*p* < 0.01) hands and the IGF-1 concentration (*p* < 0.02) were significantly higher in the EG compared with the CG. However, a comparison between the groups with respect to the change in body composition (total body fat, visceral fat area, and relative muscle mass) using the BIA and amount of change in blood pyruvate, lactate, and hsCRP concentrations using a blood analysis did not yield any significant differences between the two groups throughout the study period, based on both the PP and ITT analyses (Table 5 and Table 6), (See Appendix A).

### 3.4. Safety and Adverse Event

In the safety assessment, all subjects completed the experiment without any adverse events (AEs). There were no dropouts due to AEs. In addition, as a result of the safety evaluation through clinical laboratory test values along with physical examinations, FSO supplementation is not likely to be associated in any AEs.

## 4. Discussion

The *Crassostrea gigas* oyster is one of the most industrially important seafood in the world as a versatile food source and excellent nutrient source. They are rich in protein, minerals, and various essential amino acids [9]. GABA is a non-protein amino acid that functions as a primary inhibitory neurotransmitter for the central nervous system and contributes to motor control, GH secretion, and emotion regulation [7,20]. Oysters contain a small amount of GABA, but not enough to promote their use as a functional food. Hence, fermentation with lactic acid bacteria can enhance the GABA content in oysters [5,21].

The present study investigated whether FSO supplementation improves skeletal muscle strength in postmenopausal women. Women who had ingested FSO for 12 weeks showed a significant increase in right quadriceps muscle strength and grip force strength on both the right and left sides compared with women in the CG. The mean difference (MD) between pre- and post-trial within each group was greater in the EG than in the CG by 7.74 Nm in muscle strength and by 1.52 kg and 1.4 kg in grip force in both the right and left sides, respectively. However, the body composition showed no significant change. A previous pilot RCT with postmenopausal women also reported an increase in 60°/s knee extension and 60°/s flexion of the quadriceps muscle without a change in body composition in the EG group [15]. With the comparable results of an increase in muscle strength and muscular endurance in both studies in addition to an increase in grip strength as obtained from the present study, it may be considered that FSO is effective in improving the muscle strength of postmenopausal women via the activation of the muscular nervous system without the improvement of muscle mass.

Previous studies have reported that fermented oyster (FO) extract contains high GABA and lactate contents, which has the potential for improvements in mitochondrial metabolism and biogenesis and exercise endurance indicators in mice [13]. Furthermore, the administration of FO extract increased the insulin–like growth factor binding protein-3 (IGFBP-3) level, which mediates the action of IGF-1. Hence, the amount of change in the blood IGF-1 concentrations may induce quantitative growth in muscle improvement. The results of the current study show an increase in IGF-1 levels in both the EG and CG after 12 weeks of ingesting FSO. However, the level of IGF-1 in the EG was significantly higher, further supporting that the positive effect of FSO on muscle strength improvement is due to a functional change via the activation of the nervous system rather than a structural change.

Our study had some limitations, including a lack of biological confirmation for FSO, including environmental factors such as sleeping hours and eating habits. Furthermore, the focus of this study was only postmenopausal women; therefore the effect of FSO on people of different sexes, ages, and living conditions are still unknown. Secondly, considering the differences between the current study and the pilot study, further follow-up study time is necessary for more accurate results. Despite the above limitations, FSO can be recommended for muscle improvement.

## 5. Conclusions

In conclusion, the efficacy and safety of FSO extract for 12 weeks to increase muscle function in postmenopausal women was approved. Furthermore, there was a significant difference in IGF-1 concentration within the experimental group by FSO ingestion for 12 weeks. Hence, our findings provide evidence for the enhancing effect of FSO on muscle strength enhancement.

## Figures and Tables

**Figure 1 ijerph-19-16450-f001:**
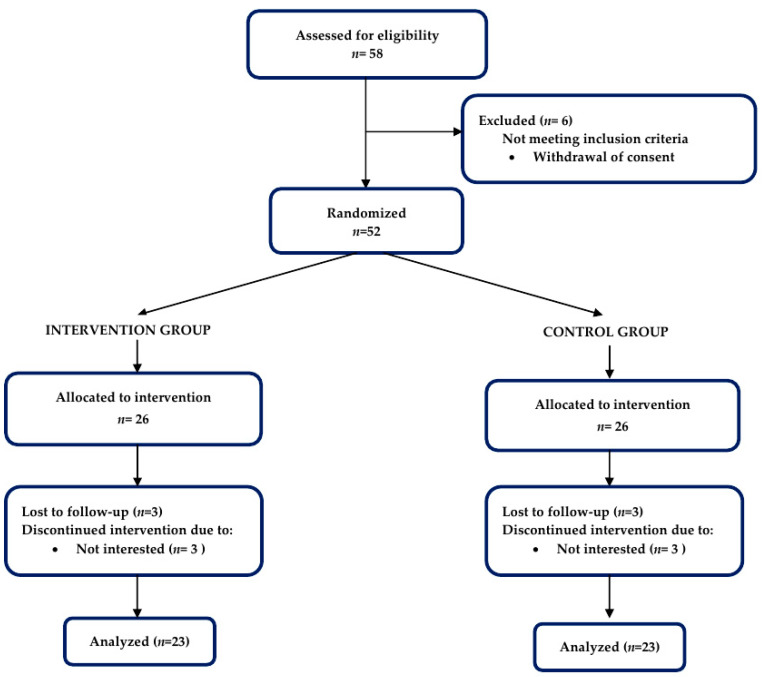
CONSORT flow diagram.

**Table 1 ijerph-19-16450-t001:** Ingredients and formulation of the test supplement and placebo supplement.

	FSO-Experiment	Placebo-Control
Common Name	Fermented Sarco Oyster extract	Dextrin
Ingredients and Contents	Fermented Sarco Oyster extract 1000 mg/day	Dextrin 1000 mg/day
Type	250 mg Capsule	Same as left
Administration Method	Take 4 capsules once a day after a meal	Same as left
Packing Unit	196 capsules in 1 bottle	Same as left
Storage Method	Room temperature storage	Same as left
Expiration Period	2 years	Same as left

**Table 2 ijerph-19-16450-t002:** Baseline demographic characteristics.

Variables	Intention-to-Treat Population			Per-Protocol Population		
	CG (*n* = 26)	EG (*n* = 26)	*p*	CG (*n* = 23)	EG (*n* = 23)	*p*
Age (years)	73.8	72.3	0.295	73.6	72.1	0.247
Weight (kg)	58.3	57.9	0.854	58.4	58.0	0.838
Height (cm)	153.5	153.4	0.949	153.5	153.6	0.930

*p* values were derived from the chi-square test; *p* values were derived from the independent *t*-test to compare between the groups.

**Table 3 ijerph-19-16450-t003:** Primary outcome comparisons between and within each group (PP population).

Variable	Observed Value			Change from the Baseline				
	CG (*n* = 23)	EG (*n* = 23)	*p* **	CG (*n* = 23)	*p* *	EG (*n* = 23)	*p* *	*p* **
60°/s knee extension peak TQ (R), Nm					0.736	6.33 ± 11.10	0.015	0.037
Visit 2	53.45 ± 12.13	54.83 ± 12.01	0.628	−1.41 ± 10.38				
Visit 4	52.05 ± 14.29	61.17 ± 15.23	0.056					
60°/s knee extension peak TQ (L), Nm					0.516	−3.74 ± 12.66	0.171	0.734
Visit 2	57.17 ± 17.03	62.00 ± 14.80	0.310	−2.26 ± 16.42				
Visit 4	54.91 ± 12.99	58.26 ± 16.88	0.455					
60°/s knee flexion peak TQ (R), Nm					0.096	1.50 ± 9.87	0.506	0.477
Visit 2	28.23 ± 9.44	32.50 ± 9.29	0.512	3.63 ± 9.46				
Visit 4	31.86 ± 11.89	34.00 ± 13.66	0.956					
60°/s knee flexion peak TQ (L), Nm					0.013	−3.74 ± 8.23	0.007	0.824
Visit 2	31.8 ± 12.4	32.3 ± 13.5	0.892	−4.26 ± 7.59				
Visit 4	27.6 ± 10.7	27.9 ± 12.3	0.919					

*p* *: Values were compared within each group; *p* **: values were compared between groups.

**Table 4 ijerph-19-16450-t004:** Primary outcome comparisons between and within each group (ITT population).

Variable	Observed Value			Change from the Baseline				
	CG (*n* = 26)	EG (*n* = 26)	*p* **	CG (*n* = 26)	*p* *	EG (*n* = 26)	*p* *	*p* **
60°/s knee extension peak TQ (R), Nm					0.985	5.77 ± 10.58	0.010	0.048
Visit 2	52.88 ± 12.15	56.27 ± 12.13	0.320	−0.04 ± 10.13				
Visit 4	52.85 ± 13.66	62.04 ± 14.35	0.023					
60°/s knee extension peak TQ (L), Nm					0.368	−4.19 ± 11.98	0.087	0.625
Visit 2	57.58 ± 16.48	62.27 ± 14.89	0.287	−2.31 ± 15.42				
Visit 4	55.27 ± 12.70	58.08 ± 16.84	0.500					
60°/s knee flexion peak TQ (R), Nm					0.118	1.50 ± 9.33	0.607	0.567
Visit 2	28.46 ± 9.24	31.00 ± 9.90	0.344	2.99 ± 9.43				
Visit 4	31.46 ± 11.86	32.50 ± 13.29	0.768					
60°/s knee flexion peak TQ (L), Nm					0.090	−0.38 ± 5.32	0.230	0.203
Visit 2	29.73 ± 13.08	31.31 ± 13.29	0.668	−2.88 ± 8.33				
Visit 4	26.85 ± 10.34	10.31 ± 12.87	0.290					

*p* *: Values were compared within each group; *p* **: values were compared between groups.

**Table 5 ijerph-19-16450-t005:** Secondary outcome comparisons between and within each group (PP population).

Variable	Observed Value			Change from the Baseline				
	CG (*n* = 23)	EG (*n* = 23)	*p* **	CG (*n* = 23)	*p* *	EG (*n* = 23)	*p* *	*p* **
Grip Force (R), Nm								
visit 2	20.26 ± 2.46	19.91 ± 2.93	0.667	1.30 ± 2.07	0.006	2.70 ± 1.47	0.000	0.011
visit 4	21.56 ± 3.56	22.61 ± 3.46	0.315					
Grip Force (L), Nm								
visit 2	19.46 ± 2.49	19.34 ± 3.65	0.901	1.48 ± 2.31	0.006	3.00 ± 2.49	0.000	0.036
visit 4	20.94 ± 3.09	22.35 ± 3.36	0.146					
Total Body Fat, %								
visit 2	37.67 ± 4.56	35.40 ± 6.54	0.177	0.70 ± 1.61	0.051	0.87 ± 1.13	0.001	0.674
visit 4	38.37 ± 4.62	36.27 ± 6.29	0.202					
Visceral Fat Area, cm^2^								
visit 2	118.4 ± 28.4	106.7 ± 39.8	0.260	9.77 ± 16.39	0.009	8.93 ± 7.96	0.000	0.827
visit 4	128.2 ± 25.9	115.7 ± 41.3	0.226					
ASM/height^2^								
visit 2	118.4 ± 28.4	106.7 ± 39.8	0.260	−0.01 ± 0.11	0.695	−0.01 ± 0.13	0.592	0.882
visit 4	128.2 ± 25.9	115.7 ± 41.3	0.226					
ASM/weight x100								
visit 2	10.45 ± 0.75	10.84 ± 1.05	0.152	−0.17 ± 0.46	0.094	−0.11 ± 6.62	0.388	0.729
visit 4	10.28 ± 0.64	10.73 ± 0.92	0.061					
IGF-1 (ng/mL)								
visit 2	104.04 ± 29.59	109.31 ± 45.91	0.646	7.13 ± 25.18	0.188	12.91 ± 23.36	0.015	0.424
visit 4	111.17 ± 26.67	122.22 ± 46.43	0.329					
hsCRP								
visit 2	0.15 ± 0.25	0.16 ± 0.36	0.924	−0.032 ± 0.167	0.371	0.05 ± 0.343	0.369	0.228
visit 4	0.12 ± 0.11	0.22 ± 0.69	0.473					
EQ-5D-3L								
visit 2	0.822 ± 0.186	0.798 ± 0.178	0.663	0.049 ± 0.156	0.144	0.061 ± 0.124	0.027 *	0.775
visit 4	0.871 ± 0.173	0.859 ± 0.157	0.812					

*p* *: Values were compared within each group; *p* **: values were compared between groups.

**Table 6 ijerph-19-16450-t006:** Secondary outcome comparisons between and within each group (ITT population).

Variable	Observed Value			Change from the Baseline				
	CG (*n* = 26)	EG (*n* = 26)	*p* **	CG (*n* = 26)	*p* *	EG (*n* = 26)	*p* *	*p* **
Grip Force (R), Nm								
visit 2	20.59 ± 2.83	19.97 ± 2.80	0.439	0.79 ± 2.94	0.192	2.58 ± 1.44	0.000	0.010
visit 4	21.38 ± 3.47	22.55 ± 3.28	0.221					
Grip Force (L), Nm								
visit 2	19.51 ± 2.79	19.34 ± 3.71	0853	1.15 ± 2.87	0.057	2.85 ± 2.39	0.000	0.025
visit 4	20.66 ± 3.12	22.19 ± 3.62	0.112					
Total Body Fat, %								
visit 2	37.63 ± 4.63	35.05 ± 6.51	0.066	0.66 ± 1.53	0.037	0.83 ± 1.09	0.001	0.674
visit 4	38.29 ± 4.65	35.88 ± 6.37	0.074					
Visceral Fat Area, cm^2^								
visit 2	121.58 ± 30.87	107.59 ± 39.33	0.160	9.54 ± 15.84	0.008	9.68 ± 7.78	0.000	0.968
visit 4	131.30 ± 28.53	117.27 ± 40.96	0163					
ASM/height^2^								
visit 2	2.59 ± 0.26	2.65 ± 0.24	0.428	−0.02 ± 0.11	0.465	−0.01 ± 0.12	0.696	0.839
visit 4	2.58 ± 0.26	2.64 ± 0.22	0.357					
ASM/weight x100								
visit 2	10.40 ± 0.74	10.83 ± 1.03	0.085	0.18 ± 0.46	0.053	−0.10 ± 0.53	0.379	0.580
visit 4	10.22 ± 0.66	10.73 ± 0.91	0.023					
IGF-1 (ng/mL)								
visit 2	105.75 ± 30.12	109.54 ± 44.92	0.733	7.03 ± 24.64	0.176	11.80 ± 22.77	0.013	0.422
visit 4	112.77 ± 27.23	122.13 ± 45.41	0.391					
hsCRP								
visit 2	0.15 ± 0.25	0.16 ± 0.35	0.830	−0.03 ± 0.16	0.371	0.05 ± 0.33	0.440	0.276
visit 4	0.12 ± 0.11	0.22 ± 0.68	0.463					
EQ-5D-3L								
visit 2	0.827 ± 0.187	0.783 ± 0.205	0.426	0.041 ± 0.148	0.169	0.066 ± 0.128	0.014 *	0.524
visit 4	0.868 ± 0.179	0.849 ± 0.164	0.688					

*p* *: Values were compared within each group; *p* **: values were compared between groups.

## Data Availability

The data that support the findings of this study are available on request from the corresponding author. The data are not publicly available due to privacy or ethical restrictions.

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
