# Peer review of "Effect of Fermented Sarco Oyster (Crassostrea gigas) Extract on Muscle Strength Enhancement in Postmenopausal Females: A Randomized, Double-Blind, Placebo-Controlled Trial"

_ijerph, 2022, doi:10.3390/ijerph192416450_

Round 1
Reviewer 1 Report
This paper is overall very well written.
The purpose of this study was to verify the effect of 12 weeks of fermented sarco oysters (FSO) intake on muscle strength in postmenopausal women with low muscle mass. In conclusion, FSO intake proved effective in strengthening muscle strength in postmenopausal women.
There are just a few clarifications and suggestions required in order to make this a high-quality paper.
First, it is necessary to indicate the catalog number, manufacturer, and country of origin of the kits used for blood biomarker analysis (Pyruvate, Lactate, hsCRP, IGF-1), and suggest the analysis methods in more detail.
Second, physical activity is a factor that can affect muscle mass and strength. Was the amount of physical activity controlled during the study period? If so, suggest it, and if not, mention it in the limitations section.
Lastly, please proofread your study. There are some grammatical errors.
Reviewer 2 Report
What does the hSCRP and IGFBP-3 mean in the line 146 and 263, respectively? Their full names should be written in the text.
It is said that ITT population is 52 (23, 23) but It seems 46 in Table 4 and Table 6. It has to be check.
The results include total body fat amount and abdominal fat amount. What are their unit? In addition, how these findings were obtained is not explained in the material and method section.
In the study, it was stated that Fermented Sarco Oyster contains GABA and that GABA will have an effect on the muscles by increasing the secretion of growth hormone and IGF-1. The authors support this thesis with the sentence "GABA is a non-protein amino acid that functions as a primary inhibitory neurotransmitter for the central nervous system and contributes to motor control, GH secretion, and emotion regulator" in lines 240-241. However, reference no.16 used here does not indicate an effect of GABA on growth hormone. Authors should check their references and find more resources to support this relationship.
